# RaceCLIP: Enhancing medical vision-language representation learning via retrieval augmented caption enrichment

## Abstract

Contrastive Language-Image Pre-training (CLIP) has demonstrated strong potential in learning transferable visual models by aligning paired images and textual descriptions. However, the quality of training data remains a significant bottleneck. In many real-world scenarios, image-text pairs are often noisy or accompanied by captions that are too short or generic to convey key visual attributes. For example, in medical imaging, most available data come from illustrative figures in public literature instead of detailed clinical reports, resulting in captions that lack the precision and context provided by expert annotations. Recent efforts to improve caption quality using Large Language Models (LLMs) have largely focused on natural images and overlook the integration of domain-specific knowledge. In this study, we propose a Retrieval-Augmented Generation (RAG) framework guided by expert semantic knowledge to enrich image captions in the medical context. We further introduce a multi-text training strategy that effectively incorporates these enriched descriptions into CLIP training. Our approach, demonstrated in the medical domain as a proof of concept, achieves state-of-the-art performances on multiple downstream tasks, highlighting its broader potential for vision-language pretraining in specialized domains. Our code is available at https://anonymous.4open.science/r/RaceCLIP-D4C5.

## 1 Introduction

Contrastive Language–Image Pretraining (CLIP) Radford et al. (2021) has shown remarkable effectiveness in learning transferable multimodal representations by aligning images and texts in a shared embedding space. CLIP leverages large-scale paired image-text datasets and uses a contrastive learning objective to associate corresponding images and captions while pushing apart mismatched pairs. This mechanism enables the model to generalize well across a wide range of downstream tasks. A major contributor to CLIP's performance is the availability of massive datasets such as LAION-5B Schuhmann et al. (2022), which are constructed by crawling image-text pairs from the whole internet. Meanwhile, it is also noticed that merely adding more web data does not always improve CLIP models Nguyen et al. (2022). Since some sources provide low-quality, noisy captions, sometimes even irrelevant with the paired images. In fact, the quality of image–text pairs often acts as a hidden bottleneck for representation learning. Noisy, incomplete, or ambiguous captions introduce weak and conflicting supervision signals, causing the contrastive objective to associate irrelevant visual features with text tokens Joshi et al. (2024). This degrades the purity of positive pairs and increases overlap with negatives, resulting in a less discriminative embedding space. Consequently, enhancing the quality of these datasets has become a pivotal aspect of training foundation models. It is also worth noting that cases of this nature have been observed in the medical field. The majority of available medical data on image-text pairs originates from illustrative figures and their captions in public medical literature, rather than case reports from hospitals, due to data regulation policies. Meanwhile, the captions of figures in the literature tend to be brief and only focus on a very small aspect of the corresponding image. Moreover, medical datasets are orders of magnitude smaller than natural datasets like LAION-5B due to patient privacy constraints which makes each medical image precious and irreplaceable. It could be unwise to discard low-quality samples in medical

Figure 1: Examples of LLM-based language rewrite vs. MLLM-based image captioning. We sample one figure from the training and validation split of ROCO dataset respectively and present the texts rewritten using LLaMa 3.2 Grattafiori et al. (2024) in yellow and the texts generated via retrieval augmented caption enrichment in green.

datasets without consequence using data filtering technique Gadre et al. (2023) commonly applied to ubiquitous web-scraped data.

To enhance the text quality of such datasets, the recaptioning of medical images using Large Language Models (LLMs) appears to be a relevant solution, given LLMs have demonstrated impressive text generation capabilities across various domains. There have been several researchers discussing how to improve the text quality by recaptioning collected images with the help of LLMs. Some focus on text rewriting, where LLMs paraphrase existing captions to introduce lexical and syntactic diversity while preserving the original meaning. These rewritten captions can be used alongside the original texts as alternative views in contrastive learning Fan et al. (2023), while in this case the visual content is discarded (see Figure 1). After the emergence of multimodal LLMs (MLLMs) exemplified by LLaVa Liu et al. (2023a), many have shifted towards generating new textual descriptions directly from visual input. These approaches aim to create richer and more informative captions by leveraging the visual contents Nguyen et al. (2023); Liu et al. (2023b). However, most existing methods rely either solely on textual input (as in language rewriting) or solely on visual input (as in image captioning), without integrating both modalities. Few works attempt to generate improved captions by jointly leveraging both the original image and its accompanying text Lai et al. (2024), which can provide complementary information and improve the accuracy and relevance of the generated descriptions. On the other hand, existing works on text augmentation always focus on natural datasets like LAION-5B, with limited attention paid to the medical domain. This gap is significant because medical image captioning presents unique challenges. Unlike natural images, where prominent objects (e.g., cats, cars) are easily recognizable, the distinction between "normal" and "abnormal" in medical imaging often exhibits minimal visual differentiation. Meanwhile, the details of pathological features (e.g., tiny tumors, slight fractures) tend to be clinically significant but visually subtle. Furthermore, accurately interpreting medical images as well as describing these nuances requires specialised knowledge that most people (and even many AI models) lack. As a result, effective captioning in the medical domain demands not only advanced multimodal reasoning but also the integration of medical knowledge.

Retrieval Augmented Generation (RAG) Lewis et al. (2020) offers a solution to these challenges by injecting domain knowledge at generation time such that MLLMs can benefit from both original caption and relevant knowledge. In this work, we propose a retrieval augmented caption enrichment framework that combines the images and text data with external expert knowledge from the United Medical Language System (UMLS) knowledge base. Additionally, we explore a multi-text contrastive loss in CLIP training to better utilize the augmented captions generated by MLLMs. Our contribution can be summarized as below:

- We propose a RAG-based medical image recaptioning framework by leveraging the latest medical MLLM, LLaVa-Med Li et al. (2023), and retrieving relevant contexts from the knowledge base UMLS using raw texts;

- We introduce a multi-text augmentation mechanism in CLIP training with one-to-many image-caption correspondences. We investigate the performances of such a strategy by comparing it with conventional CLIP which follows a strict one-to-one correspondence;

- We develop a public multimodal medical dataset $ROCO_{cap}$ by enriching captions of the original ROCO Ionescu et al. (2023) with MLLMs;

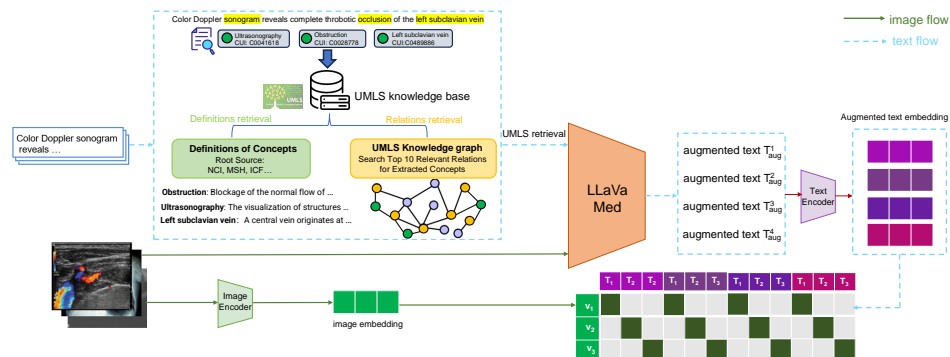

Figure 2: Overview of the proposed framework: RaceCLIP. The input captions are first used to extract medical concepts for subsequent retrieval augmented caption enrichment (shown in the dashed box). We further align the corresponding input images with all the augmented texts through a multi-text contrastive loss.

- We conduct a thorough ablation study to evaluate the benefits of enhancing medical image captioning by retrieving external knowledge (e.g., concept definitions & relations).

By integrating RAG with CLIP framework, we directly address the twin bottlenecks of caption quality, dataset scarcity in medical vision–language pretraining. It allows for transforming limited, noisy image–text pairs into a high quality, multi caption corpus. Such enriched corpus (as examplified by $ROCO_{cap}$) not only enhances image–text alignment but also yields more robust and clinically grounded visual representations.

## 2 RELATED WORKS

**Image captioning.** Developing models capable of generating captions from images has been a long-standing area of research Karpathy & Feifei (2015). In recent years, models such as BLIP Li et al. (2022) have achieved substantial advances in this domain. Building upon the combination of CLIP-based image encoder and decoder-only LLMs, rapid advancements of multimodal large language models have made LLaVa-like models Liu et al. (2023a) powerful approaches to produce more semantically enriched and informative image descriptions nowadays. In this work, our approach leverages existing image captioning systems to generate synthetic captions for medical images.

**Retrieval Augmented Generation (RAG)** was first proposed in Lewis et al. (2020) by combining a BERT-based neural retriever Karpukhin et al. (2020) with language models to enhance the text generation. After the emergence of the powerful decoder-only LLMs, such integration of retrieval modules has led to the notable development of RAG, which has become one of the most prominent methodologies in the domain of generative AI. Despite the numerous successful applications of RAG in various knowledge-intensive domains, such a useful technique is rarely adapted to image captioning tasks because a BERT-based retriever may fail to capture the subtle visual content in images Tanaka et al. (2025). However, sometimes existing metadata paired with image can also serve as queries, which can be understood by retrievers especially when dealing with medical images.

**Improving image-text datasets.** Image-text datasets crawled from the internet often suffer from issues such as low-quality textual descriptions and image-text misalignment. Many captions can be noisy, vague, and often fail to accurately reflect the visual content of the corresponding image. With this in mind, many aim to filter less informative or misaligned image-text pairs Schuhmann et al. (2022); Gadre et al. (2023). However a significant amount of data may be discarded, despite the presence of rich semantic content in certain images. Another method of data refinement is data recaptioning, leveraging advanced LLMs to generate new captions. Whilst some focus on language rewriting, for example in Fan et al. (2023); Bahadir et al. (2025), this process is rendered meaningless if the quality of the original text is poor. Some aim to recaption images purely based on visual content, while neglecting to consider the many meaningful contexts that can be retrieved and added to the prompt for better generation performance.

## 3 METHODOLOGY

In this section, we detail our methodological contribution called RaceCLIP (**R**etrieval **a**ugmented **c**aption **e**nrichment-based **C**ontrastive **L**anguage **I**mage **P**retraining), illustrated in Figure 2. It is composed of two components: (1) a retrieval augmented caption enrichment module that improves the quality and diversity of description for medical images, and (2) a CLIP framework paired with a multi-text contrastive loss which allows the image encoder to synchronously learn from multiple (diverse) captions.

### 3.1 UNITED MEDICAL LANGUAGE SYSTEM (UMLS)

As input, we have an initial dataset named $X$ which contains pairs of images and captions. We discuss hereinafter how we can generate new captions from existing data guided by domain knowledge. To this end, we aim to retrieve external medical knowledge from the UMLS (initially introduced in Bodenreider (2004)), a comprehensive biomedical vocabulary covering a wide range of medical concepts. The UMLS employs Concept Unique Identifiers (CUIs) to distinguish medical concepts, where each CUI is a string beginning with 'C' and ending with seven numerical digits (e.g., C0041618 refers to the concept *ultrasonography*). Besides, the UMLS also includes the definitions of medical concepts as well as a huge semantic knowledge graph (KG) which links terms from over 200 vocabularies and enables semantic interoperability across terminologies. In this step, our goal is to retrieve the accurate definitions of medical concepts and the most relevant relationships to augment the medical context for better image captioning performance.

### 3.2 RETRIEVAL AUGMENTED IMAGE RECAPTIONING

Firstly, medical concepts hidden in the original captions (belonging to the input $X$ dataset) are extracted using Scispacy Neumann et al. (2019). Following this, each concept is matched to its corresponding CUI in the UMLS. We then retrieve the definitions and contextual relationships corresponding to each concept through CUIs. This process augments the prompt for caption generation. And such additional knowledge will help MLLM to better understand the specific medical context of each image. For the definition retrieval, each CUI may correspond to more than one definition in the UMLS from different sources such as the "National Cancer Institute (NCI)" and the "Medical Subject Headings (MSH)". By default, we prioritize English language definitions and retrieve the shortest one with each CUI as a query. It needs to be noticed that some CUIs may not be defined in the above sources, in such cases we retrieve the definition from Wikipedia instead.

In order to retrieve relations, we make use of the entire knowledge graph from the metathesaurus of the UMLS. This connects 4.48 million CUIs through 75 million relationships. For each extracted CUI, we regard all the directly connected nodes and the associated edges as candidate relations. The knowledge graph (KG) is decomposed into a set of knowledge triplets $S = \{(h_i, l_i, t_i) | i = 1, \ldots, N_r\}$ where $N_r$ is the number of edges in the KG. The symbols $h_i$, $l_i$ and $t_i$ refer to the head node, the relation and the tail node, respectively. We then express each relation as text by connecting the head node and the tail node using the specified relation type (e.g., triplet like (C0005682: Urinary bladder, procedure_cite_of, C0200002: Complex cystometrogram) can be rewritten as follows: "The urinary bladder is the procedure site of complex cystometrograms – calibrated electronic equipment"). For each candidate relation rewritten as text, we employ PubMedBERT Gu et al. (2021), a domain-specific language model pretrained on the PubMed literature for biomedical natural language processing, to extract the textual representation of candidate relations. Meanwhile, with the original caption as the query, we also extract the embeddings for queries using the same text encoder. Finally, by comparing the cosine similarities between the query and candidates, we retrieve the top 10 relevant relations for prompt augmentation.

After retrieving the definitions and relevant relations, we combine such contextual knowledge with the instruction prompts (e.g., "provide a clinical description of this medical image") to generate new image captions using LLaVa-Med Li et al. (2023), a domain-adapted version of LLaVA fine-tuned for medical tasks. With each image-caption pair as input, we generate four augmented texts $T_{aug}^1, T_{aug}^2, T_{aug}^3, T_{aug}^4$ for further vision-language contrastive learning, depending on whether retrieve definitions/relations or not (see Table 3). Such a retrieval augmented image recaptioning method leads to a novel dataset $X_{cap}$, extending the initial image-caption dataset $X$ with enriched

captions. The proposed approach is naturally extensible to more than four augmented texts depending on the needs.

### 3.3 CLIP WITH MULTI-TEXT CONTRASTIVE LOSS

After the retrieval augmented caption enrichment, we can then feed the images paired with multiple augmented texts to the vision language model for enhanced contrastive representation learning. To this end, we extend and generalize the architecture of the CLIP framework as shown in Figure 2.

Our dual stream architecture is composed of a Vision Transformer-based image encoder $H_{visual}$ and a GPT-based text encoder $H_{text}$. For the multi-modal input, each image $I$ is paired with multiple texts $(T_{aug}^1, T_{aug}^2, T_{aug}^3, T_{aug}^4)$ generated by MLLMs. We encode them separately with the corresponding encoder:

$$\mathbf{v} = H_{visual}(I)$$

and

$$\mathbf{t}^1, \mathbf{t}^2, \mathbf{t}^3, \mathbf{t}^4 = H_{\text{text}}(T_{\text{aug}}^1, T_{\text{aug}}^2, T_{\text{aug}}^3, T_{\text{aug}}^4)$$

where $\mathbf{v}$ are the embeddings of the images and $\mathbf{t}^1, \mathbf{t}^2, \mathbf{t}^3, \mathbf{t}^4$ correspond to the embeddings of the different augmented texts that are generated based on different retrieval result using LLaVa-Med.

In order to benefit from various augmented texts paired with each image, we propose to align the visual representation with multiple textual representations simultaneously. Similarly to the design of the InfoNCE loss, we design the multi-text contrastive loss as follows: $L_{MT} = L^{i \to t} + L^{t \to i}$ where $L^{t \to i}$ and $L^{i \to t}$ model the cross-entropy loss for optimizing the cross-modality prediction. The one-to-one contrastive loss is then extended to a one-to-many format. Specifically, for image-to-text prediction, we compute the loss as described hereinafter.

With a batch size equal to $N$, given an image paired with $M$ augmented captions ranging from $1, 2, \ldots, N$, the term $L^{i \to t}$ can be expressed as:

$$L_{i \to t} = -\frac{1}{M} \sum_{i=1}^{N} \sum_{j=1}^{M} \log \frac{\exp(\left\langle \mathbf{v}_i, \mathbf{t}_i^j \right\rangle / \tau)}{\sum_{k=1}^{N} \exp(\left\langle \mathbf{v}_i, \mathbf{t}_k^j \right\rangle / \tau)} \tag{1}$$

where $\left\langle \mathbf{v}_i, \mathbf{t}_i^j \right\rangle$ denotes the cosine similarity of the $i$-th pair of visual and textual embedding, $j$ represents the $j$-th augmented text generated by LLaVa-Med, and $\tau$ is a learnable temperature for the contrastive loss.

In practice, all the generated texts have only one corresponding image, the text-to-image contrastive loss $L^{t \to i}$ follows the same design as CLIP training which can be written as:

$$L_{t \to i} = -\frac{1}{M} \sum_{i=1}^{N} \sum_{j=1}^{M} \log \frac{\exp(\left\langle \mathbf{t}_i^j, \mathbf{v}_i \right\rangle / \tau)}{\sum_{k=1}^{N} \exp(\left\langle \mathbf{t}_i^j, \mathbf{v}_k \right\rangle / \tau)} \tag{2}$$

Rather than aligning randomly transformed images with the same unaltered texts in each iteration, optimizing CLIP with multiple captions at the same time helps to avoid overfitting during training.

## 4 EXPERIMENTAL STUDY

### 4.1 DATASETS

**ROCO** is a large-scale medical image-text pair dataset introduced as part of the ImageCLEF 2023 challenge Ionescu et al. (2023). It comprises 60,918/10,437/10,473 image-text pairs in the training, validation and test splits. All image-text pairs are sourced from the figures of PubMed articles with corresponding captions, encompassing a diverse range of imaging modalities and anatomical regions. Additionally, each image is associated with a set of CUIs extracted from its caption. Note that ROCO serves as the primary dataset in this study: we enrich the captions from ROCO using LLaVa-Med leading to the $ROCO_{cap}$ dataset and optimize our model on the training and validation

splits of $ROCO_{cap}$, the test set of ROCO is involved in downstream image retrieval and image classification tasks. The created $ROCO_{cap}$ dataset will be made available to the community upon acceptance.

**MEDICAT** Subramanian et al. (2020) comprises 217k images extracted from biomedical articles, with 75% of the images being composite figures. Each image is accompanied by its caption and the corresponding inline reference within the paper. We randomly sample 8000 image-caption pairs for cross modality retrieval evaluation.

**MURA** Rajpurkar et al. (2017) is a X-ray dataset of musculoskeletal radiographs covering seven anatomical regions like elbow, shoulder and forearm. For the evaluation, we conduct image retrieval experiments on its test set using the body part labels. In addition, we also visualize the learned image embeddings as part of qualitative evaluation.

**IRMA** Lehmann et al. (2003) is an X-ray image dataset of 14k images of different parts of the human body. Every image is associated with a 13-digit hierarchical IRMA code of the following form (IRMA: TTTT - DDD - AAA - BBB), where T, D, A, and B denote a coding or sub-coding digit of the technical, directional, anatomical, and biological axis, respectively. For the zero-shot image classification tasks, we select a subset of 9 classes paired with 200 samples per class from the test set using labels of directions and the anatomy.

## 4.2 IMPLEMENTATION DETAILS

For the MLLM-assisted caption enrichment, we retrieved the definitions and relationships from the UMLS 2025AA version and we chose LLaVa-Med-v1.5-7B as the image captioner. The context retrieval was performed offline and it took around 90 hours to recaption ROCO on a V100 GPU.

For the CLIP architecture, we chose ViT-B/16 as the image encoder and kept the GPT-2 as the text encoder. We initialized both encoders with OpenAI CLIP weights pretrained on WIT (WebImage-Text) dataset. The model was then fine-tuned using the official training and validation splits of ROCO with the enriched texts via retrieval augmented generation on a V100 GPU for 15 epochs with a batch size of 32. We chose Adam for the optimizer paired with a learning rate of 3e-6. Source codes are included in the zip file of supplementary materials and the extended $ROCO_{cap}$ dataset will be shared via Huggingface if this paper get accepted.

## 4.3 BASELINES

**CLIP** Radford et al. (2021) is a vision-language model that learns to connect images and natural language through contrastive learning. Its core idea is to understand both images and text by aligning them in a shared embedding space.

**MedCLIP** Wang et al. (2022) is a CLIP framework tailored for medical imaging which aims to learn joint visual-language representations for medical images and reports, without requiring paired image-text data. This is crucial since paired datasets are limited in medical domains.

**LaCLIP** Fan et al. (2023) is a language augmented CLIP framework specialized on text augmentation. It applies language rewrite to the original caption using LLaMa Touvron et al. (2023) to increase the diversity of texts. The core idea is to prevent aligning various randomly transformed images with the unchanged text in each iteration.

**VeCLIP** Lai et al. (2024) is another language augmented CLIP framework benefiting from the visual-enriched captions. The authors noticed the shortcoming of language rewrite which fails to return a good caption if the given text is poor. They generate detailed captions by exploiting the visual content using LLaVa Liu et al. (2023a) as an image captioner. Another LLM is used to fuse the original captions and the generated captions.

## 4.4 QUANTITATIVE EVALUATION

To quantitatively evaluate the quality of the visual and textual representations, we involve them in image-to-image retrieval, cross modality retrieval and image classification tasks in a zero-shot setting.

| Methods | ROCO CUI@K | | | Custom retrieval dataset P@K | | | | | | | | | MURA P@K | | |
|---------|------|------|------|------|------|------|------|------|------|------|------|------|------|------|------|
| | @5 | @10 | @50 | @5 | @10 | @30 | @5 | @10 | @30 | @5 | @10 | @30 | @5 | @10 | @30 |
| | | | | Modality | | | Organ | | | Modality/Organ | | | Organ | | |
| Random | 3.12 | 4.20 | 5.37 | 20.00 | 20.00 | 20.00 | 12.50 | 12.50 | 12.50 | 14.28 | 14.28 | 14.28 | 14.28 | 14.28 | 14.28 |
| CLIP | 39.85 | 40.84 | 44.86 | 94.85 | 93.14 | 91.02 | 84.70 | 81.07 | 75.27 | 89.67 | 88.02 | 84.33 | 69.96 | 57.82 | 50.55 |
| MedCLIP | 39.92 | 40.98 | 44.78 | 95.31 | 93.28 | 91.66 | 84.81 | 81.02 | 75.51 | 89.92 | 88.91 | 84.73 | 73.49 | 59.21 | 56.49 |
| LaCLIP | 40.23 | 41.49 | 45.58 | 95.48 | 94.08 | 91.58 | 85.28 | 82.21 | 77.12 | 90.33 | 89.09 | 85.23 | 71.03 | 59.02 | 53.61 |
| VeCLIP | 40.89 | 41.83 | 46.19 | 95.56 | 94.41 | 92.35 | 85.59 | 83.01 | 77.31 | 90.28 | 89.25 | 85.41 | 73.41 | 58.38 | 56.62 |
| RaceCLIP | **42.27** | **43.52** | **48.32** | **96.59** | **96.59** | **95.48** | **86.65** | **83.98** | **79.70** | **91.81** | **91.04** | **89.09** | **82.48** | **69.54** | **75.35** |

Table 1: Zero-shot image-to-image retrieval results using ViT-B/16 as the image encoder. Left columns denote CUI@K on the ROCO test sets, mid columns denote P@K on the custom (ROCO) retrieval dataset, right columns denote P@K on the MURA dataset. Random refers to the result of a random guess.

**Zero-shot image-to-image retrieval.** In the following experiments, we perform image retrieval as follow: (1) Firstly we extract image representations using our image encoder; (2) Then, with each image as a query, we rank all the other candidate images based on the cosine similarity between the query and candidates; (3) Finally we compare the performance using the top $K$ retrieved images. We report the results using these metrics:

- **CUI@K** is an image retrieval metric proposed in Sérieys et al. (2022) to measure the retrieval performance using CUIs. Specifically, for each query image, we first compute the Jaccard index between the set of CUIs paired with that image and the set of CUIs paired with all candidate images. Then, we rank all candidate images in descending order using these Jaccard scores and treat this ranking as the ground truth for the image-to-image retrieval task. Then, we compute the CUI@K metric in a Normalized Discounted Cumulative Gain (NDCG) manner by comparing the predicted ranking result and the ground truth;

- **Precision@K** is calculated by dividing the number of relevant samples among the top $K$ by $K$. Here we create a test set for retrieval customized from ROCO (referred as "custom retrieval dataset") with the following steps: First, we generate potential labels related to imaging modalities and organs using the semantic types of CUIs. More precisely, we selected images paired with 5 modalities (e.g., "ultrasound", "tomography") according to semantic type T060 "diagnostic procedure", and images paired with 10 organs (e.g., "teeth", "lung", "liver") according to semantic type T023 "body part, organ". We can then evaluate three image retrieval tasks (i.e. "Modality", "Organ", "Modality & Organ") based on the precision@K metric.

Table 1 presents the comparative results for image retrieval. We observe that our method outperforms LaCLIP and VeCLIP which indicates our domain specific retrieval augmented image recaptioning framework generates better captions compared with other LLMs without retrieval based augmentation. Moreover, our model shows superiority to CLIP with notable improvements which confirms the importance of the high-quality texts during the training.

**Zero-shot cross modality retrieval.** To evaluate the performance of both encoders, we first extract visual and textual representations for the image-text pairs using our model. Then with each image/text as query respectively, we perform Image-to-Text (I2T) retrieval and Text-to-Image (T2I) retrieval tasks using the image-text similarity in the shared embedding space. We randomly sample 8000 image-caption pairs from the MEDICAT dataset and we report the results in terms of Recall@k as R@1, R@5, and R@10.

We summarize the cross modality retrieval results in Table 2b. Our method shows consistent improvements across Recall@k metrics in both I2T and T2I retrieval tasks for the MEDICAT dataset.

**Zero-shot image classification.** Similar to the evaluation process in Radford et al. (2021), we conduct zero-shot image classification tasks for X-ray datasets as follow: (1) Firstly, with each label of body parts, we create a text prompt using following template `"A radiograph of {label}"`, after that, we extract the textual representations of prompts for all possible labels; (2) Secondly, we extract the visual representation for the radiograph using our image encoder and compute the image-prompt similarity for all possible labels; (3) Finally, we classify the radiographs as the class according to the image-prompt matching probability from the highest to the lowest. We report the classification accuracy in Table 2b. The results showcase that our method outperforms other base-

| Methods | MURA | | IRMA | |
|---|---|---|---|---|
| | Top-1 | Top-5 | Top-1 | Top-5 |
| Random | 14.28 | 71.43 | 11.11 | 55.56 |
| CLIP | 50.28 | 96.58 | 54.44 | 95.83 |
| MedCLIP | 53.43 | 97.25 | 54.55 | 95.91 |
| LaCLIP | 51.82 | 96.91 | **54.83** | 96.77 |
| VeCLIP | 53.36 | 97.14 | 54.71 | 97.28 |
| RaceCLIP | **60.73** | **98.40** | 54.77 | **99.78** |

| Methods | MEDICAT | | | | | |
|---|---|---|---|---|---|---|
| | I2T | | | T2I | | |
| | R@1 | R@5 | R@10 | R@1 | R@5 | R@10 |
| Random | 0.01 | 0.06 | 0.12 | 0.01 | 0.06 | 0.12 |
| CLIP | 17.50 | 35.89 | 46.75 | 16.36 | 35.25 | 45.56 |
| MedCLIP | 17.62 | 35.96 | 46.68 | 16.48 | 35.37 | 45.52 |
| LaCLIP | 18.23 | 36.68 | 47.81 | 17.05 | 36.37 | 46.81 |
| VeCLIP | 18.76 | 38.21 | 48.33 | 17.61 | 37.69 | 47.22 |
| RaceCLIP | **20.12** | **40.03** | **50.59** | **19.88** | **39.77** | **48.62** |

(a) Zero-shot image classification accuracy on the X-ray MURA and IRMA datasets. Random refers to the result of a random guess.

(b) Zero-shot cross-modality retrieval results on the MEDICAT dataset. Random refers to the result of a random retrieval.

Table 2: Zero-shot evaluation results for (a) Classification tasks. (b) Cross modality retrieval tasks.

| text | RAG | | ROCO | | | Custom retrieval dataset | | | | | | | | | MURA | | |
|---|---|---|---|---|---|---|---|---|---|---|---|---|---|---|---|---|---|
| | def | rel | CUI@K | | | P@K | | | | | | | | | P@K | | |
| | | | @5 | @10 | @50 | @5 | @10 | @30 | @5 | @10 | @30 | @5 | @10 | @30 | @5 | @10 | @30 |
| | | | | | | Modality | | | Organ | | | Modality/Organ | | | Organ | | |
| $T_{aug}^1$ | | | 39.72 | 40.85 | 44.71 | 95.11 | 93.61 | 91.78 | 83.81 | 81.47 | 77.12 | 90.40 | 89.23 | 86.40 | 69.88 | 57.95 | 50.62 |
| $T_{aug}^2$ | | ✓ | 41.77 | 42.74 | 46.89 | 96.34 | 95.47 | 94.42 | 86.44 | 83.87 | 79.81 | 91.36 | 90.77 | 89.28 | 77.63 | 64.57 | 71.29 |
| $T_{aug}^3$ | ✓ | | 41.85 | 42.98 | 47.12 | 96.36 | 95.14 | 93.75 | 85.79 | 83.49 | 79.50 | 91.38 | 91.58 | 89.98 | 78.24 | 65.89 | 72.12 |
| $T_{aug}^4$ | ✓ | ✓ | 42.01 | 43.12 | 47.77 | 96.47 | 95.58 | 94.26 | 85.88 | 83.79 | 79.62 | 91.35 | 90.99 | 89.57 | 79.82 | 67.92 | 74.01 |
| all | ✓ | ✓ | **42.27** | **43.52** | **48.32** | **96.59** | **96.59** | **95.48** | **86.65** | **83.98** | **79.90** | **91.81** | **91.04** | **90.09** | **82.48** | **69.54** | **75.35** |

Table 3: Ablation study for the RAG framework and the multi-text alignment strategy (def and rel refer to concept definitions and semantic relations from the UMLS). Best scores are indicated in bold font and we use blue underlines to highlight the best performance for each augmented text.

| image captioner | ROCO | | | Custom retrieval dataset | | | | | | | | | MURA | | |
|---|---|---|---|---|---|---|---|---|---|---|---|---|---|---|---|
| | CUI@K | | | P@K | | | | | | | | | P@K | | |
| | @5 | @10 | @50 | @5 | @10 | @30 | @5 | @10 | @30 | @5 | @10 | @30 | @5 | @10 | @30 |
| | | | | Modality | | | Organ | | | Modality/Organ | | | Organ | | |
| LLaVa | 39.95 | 40.79 | 44.84 | 95.24 | 93.75 | 91.91 | 83.89 | 81.62 | 77.03 | 90.72 | 89.46 | 86.87 | 70.12 | 58.88 | 50.97 |
| Qwen2.5-VL | 41.95 | 42.96 | 47.43 | 96.21 | 96.03 | 94.45 | 85.91 | 83.64 | **79.85** | 90.98 | 90.63 | 88.89 | 80.81 | 67.91 | 74.21 |
| LLaVa-Med | **42.27** | **43.52** | **48.32** | **96.59** | **96.59** | **95.48** | **86.65** | **83.98** | 79.70 | **91.81** | **91.04** | **89.09** | **82.48** | **69.54** | **75.35** |

Table 4: Ablation study for choosing various MLLMs as the image captioner for the proposed RaceCLIP framework.

lines except for top-1 scores on the IRMA dataset. Since IRMA is the only dataset with directional labels in this study, this phenomenon may stem from the hallucination of MLLMs, e.g., LLaVa-based models can generate some false directional description (sagittal/frontal/horizontal) in the augmented texts. This suggests that LLaVa-based image captioning may not always outperform LLama-based language rewrite in terms of reliability.

## 4.5 QUALITATIVE EVALUATION

In this section, we study the visual embeddings of the MURA dataset and image-text similarities for the recaptioned ROCO using LaCLIP, VeCLIP and our method.

**Visual embedding interpretation.** We visualize the image embeddings on the MURA dataset using a t-SNE (see Figure 3a). We observe that CLIP leads to messy plot and entanglement between each cluster. Such a visualization highlights that our method leads to well-separable features. Besides, considering relatedness between classes, we can see that images with similar labels are located close to each other (e.g., wrist and hand labelled in purple and blue) and images with relatively dissimilar labels are located far apart (e.g., shoulder labelled in brown are located far from others except humerus) for our method.

**Image-text similarity visualization.** To intuitively evaluate the performance of the retrieval augmented caption enrichment module, we plot and provide in Figure 3b the image-text similarity

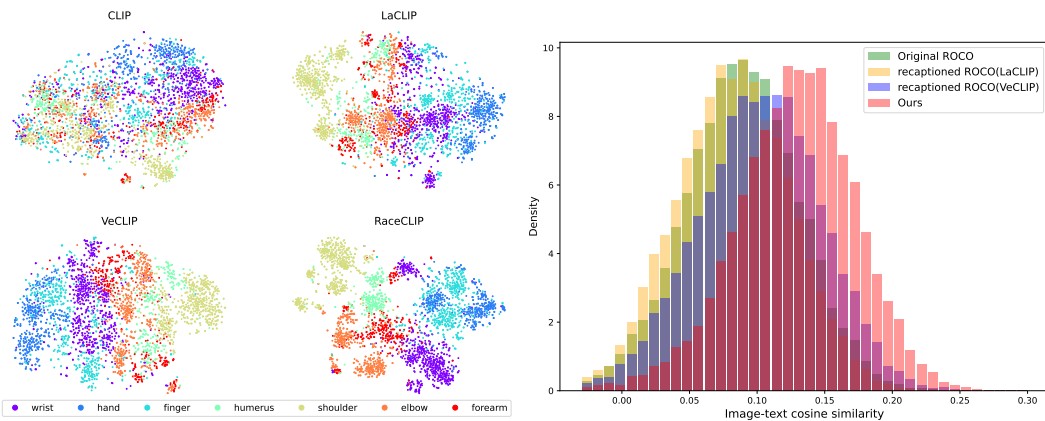

(a) T-SNE plots of the visual embeddings on the MURA dataset.

(b) Cosine similarity distribution of images and captions. Our method improves the average matching degree of image-caption pairs.

Figure 3: Visualization results for the qualitative evaluation. (a) T-SNE of embeddings on MURA. (b) Cosine similarity distribution between images and captions.

distribution of the recaptioned ROCO datasets using our method and other baseline methods. It can be observed that our method increases the average matching degrees between image-caption pairs compared with the original ROCO dataset and the ROCO dataset recaptioned using LaCLIP and VeCLIP.

### 4.6 ABLATION STUDY

To prove the effectiveness of our RAG-based framework, we propose to train CLIP by aligning ROCO images with augmented texts generated with and without knowledge retrieval from the UMLS in a conventional one-to-one format (first four rows in Table 3). A comparison between the first row and subsequent rows reveals that the retrieved concept definitions and semantic relations improve the quality of generated captions. Furthermore, to demonstrate the effectiveness of our multi-text contrastive learning strategy, we also report the performances of our RaceCLIP proposal in the final row of Table 3 by synchronously aligning ROCO images with all the augmented texts $T_{aug}^1, T_{aug}^2, T_{aug}^3, T_{aug}^4$. By comparing the performance of the final row and the first four rows, we note that aligning ROCO images with all the generated texts yields better results than aligning them with any single caption variant described above.

Additionally, to investigate the effectiveness of LLaVa-Med as the image captioner, we conduct another experiment by replacing it with other 7B open-source MLLMs. We can see in Table 4 that LLaVa-Med outperform LLaVa and Qwen2.5-VL due to a successful medical domain adaption.

### 4.7 CONCLUSION

In this work, we propose the RaceCLIP framework which consists of (1) a MLLM-assisted retrieval augmented image recaptioning module to generate better descriptions for medical images by integrating prompts with expert knowledge retrieved from the UMLS and (2) a CLIP framework with multi-text alignment strategy. Our results show that the retrieval augmented image recaptioning module helps to improve the textual quality of a medical dataset and yield competitive performances against other state-of-the-art CLIP frameworks with text augmentation. Meanwhile, we believe that our contribution can be generic and extended to other domains with the access to other knowledge bases like wikitionary and WordNet Fellbaum (2010).

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

## A  APPENDIX

### LIMITATIONS

Nevertheless, RaceCLIP still presents some limitations offering longer-term research perspectives. First, our framework inevitably comes with higher computational cost due to MLLM-assisted dataset recaptioning (360 GPU hours of inferencing using LLaVa-Med for ROCO) and the multi-text alignment contrastive loss (3% more time for training in each epoch compared with CLIP). Meanwhile, hallucinations of LLMs still remain an unsolved problem today. Though our RAG-based image recaptioning module can alleviate the hallucination to some extent by integrating factual knowledge in the context window, the reliability of synthetic medical captions needs to be verified by well-trained professionals. Furthermore, other fact-checking pipelines could be considered as part of post-generation verification for future works.

### ETHICS STATEMENT

This work addresses vision-language representation learning in the medical domain. We acknowledge the sensitive nature of medical data and the ethical considerations that come with it. The dataset used in this study, ROCO, is sourced from publicly available figures in PubMed articles. Patient privacy is of utmost importance, and our work strictly adheres to data usage policies. We also recognize that the application of such models in a clinical setting would require careful validation and adherence to ethical guidelines to ensure patient safety and avoid potential biases. The authors have read and adhere to the ICLR Code of Ethics.

### REPRODUCIBILITY STATEMENT

The source code for our methodology is included in the abstract and the supplementary materials. The extended $ROCO_{cap}$ dataset, created by our retrieval-augmented generation framework, will be made publicly available upon acceptance of this paper. The methodology section (Section 3) of this paper provides a detailed description of the model architecture, data processing steps, and training procedure to ensure the reproducibility of our results.

### LLM USAGE STATEMENT

This paper uses Large Language Models (LLMs), specifically LLaVa-Med, as a core component of the research methodology. The LLM was used to generate new captions to augment the original dataset and enhance the training process. LLMs were not used for the generation of the paper's text itself.

