# OpenReview forum: "RaceCLIP: Enhancing medical vision-language representation learning via retrieval augmented caption enrichment"
_ICLR.cc/2026/Conference — Submitted to ICLR 2026_

### Official Review · Reviewer_kptq · 2025-10-28

**Soundness:** 3
**Presentation:** 3
**Contribution:** 2
**Rating:** 2
**Confidence:** 4

**Summary:**

This paper proposes RaceCLIP, a framework that combines Retrieval-Augmented Generation (RAG) with CLIP training for medical vision-language representation learning. The authors use LLaVa-Med with UMLS knowledge retrieval to generate enriched captions for the ROCO dataset (81K image-text pairs), creating ROCOcap, and introduce a multi-text contrastive loss to align images with multiple augmented captions simultaneously. The method is evaluated on image retrieval and zero-shot classification tasks across multiple medical imaging datasets, showing improvements over baselines including CLIP, MedCLIP, LaCLIP, and VeCLIP.

**Strengths:**

1. Systematic engineering approach. The paper demonstrates careful integration of RAG with medical image captioning, thoughtfully combining UMLS knowledge (definitions and relations) with multimodal LLMs to generate enriched descriptions.
2. Comprehensive experimental validation. The authors provide detailed ablation studies (Table 3) showing the contribution of different retrieval components and evaluate their method across diverse downstream tasks including image-to-image retrieval, cross-modality retrieval, and zero-shot classification on multiple datasets (ROCO, MEDICAT, MURA, IRMA).
3. Reproducibility commitment. The authors promise to release both the source code and the ROCOcap dataset upon acceptance, which would benefit the research community.
4. Domain-specific knowledge integration. The paper demonstrates how to systematically incorporate expert semantic knowledge (UMLS) into the caption generation process, which is particularly relevant for specialized domains like medical imaging.

**Weaknesses:**

1. The paper claims medical datasets are "orders of magnitude smaller" than natural image datasets due to privacy constraints, but this is demonstrably false as of 2025. Multiple large-scale medical image-text datasets now exist, including BIOMEDICA (24M+ pairs), PMC-OA (1.6M pairs), PMC-15M (15M pairs). The ROCO dataset (81K pairs) used in this paper represents only very few portion of available medical imaging data.

2. The authors train exclusively on ROCO (81K low-quality figure captions) without providing any scientific justification for why they ignore vastly larger and higher-quality alternatives like BIOMEDICA or PMC-15M. Spending 360 GPU hours to generate 81K synthetic captions (which may contain hallucinations) instead of using 24M+ real captions is questionable, and the paper provides zero rationale beyond implicit convenience.

3. The paper compares against methods from 2021-2024 but ignores modern medical CLIP models trained on large-scale data (PMC-OA, PMC-15M, BIOMEDICA). A fair comparison requires evaluating whether RAG on 81K samples outperforms simple scaling to millions of real samples, and without comparing "ROCO (81K) + RAG" versus "BIOMEDICA (24M) baseline," we cannot assess whether RAG adds value or if data scaling is simply more effective.

4. The paper provides no quantitative hallucination analysis, no expert validation of generated captions, and no comparison of error rates between synthetic and real clinical reports.

5. Limited novelty as incremental combination of existing techniques like RAG, LLM-based recaptioning from LaCLIP/VeCLIP, multi-text contrastive learning.

6. The paper provides no cost-benefit analysis comparing performance per GPU hour against simply training on larger existing datasets, and cherry-picks metrics while dismissing cases where RaceCLIP underperforms (e.g., IRMA Top-1: 54.77 vs LaCLIP 54.83).

7. All experiments are conducted only on small datasets (ROCO test 10K, MEDICAT 8K) without evaluation on large-scale benchmarks, and it's unclear whether the RAG approach scales to millions of images or how the 3% training overhead increases with more augmented texts.

**Questions:**

See weakness

---

### Official Review · Reviewer_Q3WE · 2025-10-28

**Soundness:** 2
**Presentation:** 2
**Contribution:** 2
**Rating:** 2
**Confidence:** 4

**Summary:**

RACECLIP introduces a UMLS-based retrieval mechanism combined with multi-text contrastive learning to address well-known limitations of standard CLIP models in the medical domain, particularly their limited domain knowledge and reliance on noisy captions. The method shows promise in improving medical understanding and overall performance.

However, the pipeline lacks sufficient rigor and transparency in several key stages. The concept extraction and CUI matching steps are not clearly explained, the clinical validity of retrieved relationships is not systematically verified, and the risks of semantic drift or hallucination in generated captions are not thoroughly analyzed. Strengthening these aspects with more detailed evaluation and failure case analysis would enhance the soundness and practical applicability of the approach.

**Strengths:**

The authors introduce RACECLIP, which integrates UMLS-based retrieval and a multi-text contrastive learning strategy to address known limitations of conventional CLIP models in medical settings, such as limited medical knowledge and noisy captions.

**Weaknesses:**

Overall, more rigorous experiments and clearer step-by-step pipeline descriptions are needed.

1.	Insufficient clarity in concept extraction and CUI matching: The ScispaCy-based mapping process lacks concrete details on similarity metrics, thresholds, and synonym resolution strategies. Without quantitative error analysis, it is difficult to assess how mapping errors propagate into downstream components.

2.	Limited validation of UMLS-based relationship retrieval: Relations not explicitly encoded in UMLS are filtered with PubMedBERT, yet there is no systematic validation that these inferred relations align with clinical context, raising concerns about semantic drift. Human review and more thorough analyses are needed.

3.	Inadequate verification of generated captions: Errors introduced in retrieval can yield semantic distortion or hallucination during caption generation. While this possibility is acknowledged, the paper does not provide deeper failure analyses or concrete mitigation mechanisms.

**Questions:**

n/a

---

### Official Review · Reviewer_TYM2 · 2025-10-31

**Soundness:** 3
**Presentation:** 2
**Contribution:** 2
**Rating:** 4
**Confidence:** 3

**Summary:**

This paper proposes RaceCLIP, a medical vision-language framework using retrieval-augmented generation (RAG) and domain knowledge (UMLS) to enrich image captions and train CLIP with multi-text contrastive learning. Experiments on multiple medical datasets show consistent gains over existing CLIP baselines.

**Strengths:**

1. Integration of RAG with domain-specific knowledge is technically sound.
2. Clear motivation on improving noisy medical captions, addresses the common issue of poor-quality captions in medical datasets.
3. Releases an enriched ROCO dataset for community use.
4. Private code sharing is considered as a merit.

**Weaknesses:**

1. Introduction is somewhat messy wth very long paragraph, clearer structure on problem, challenge, and motivation needed.
2. Incremental combination of RAG and MLLM leads to limited methodological novelty.
3. Experiments are relatively narrow with no comparison to recent RAG-based works, and omits non-medical domains, weakening generalization claims.

**Questions:**

Can the authors clarify the key novelty and how it differs from recent RAG-based medical VL frameworks (e.g., HeteroRAG, MMED-RAG)?

---

### Official Review · Reviewer_kdfY · 2025-11-03

**Soundness:** 3
**Presentation:** 2
**Contribution:** 2
**Rating:** 2
**Confidence:** 4

**Summary:**

This paper investigates a technique for harnessing synthetic data for data augmentation to train CLIP models in the medical domain.
In particular, it uses RAG over knowledge resources (UMLS) for data augmentation to obtain additional textual descriptions.
This extended data is then used to train an improved CLIP model.

**Strengths:**

- Improvements over some prior models

**Weaknesses:**

- The motivation provided in the introduction does not align that well with the developed solution. The authors rightly point out that "the distinction between “normal” and “abnormal” in medical imaging often exhibits minimal visual differentiation." However, it is not clear why data augmention from a knowledge store such as UMLS is the most appropriate solution in this case.

- The paper also presents a loss functon for handling multiple texts per image in Section 3.3. Howvever, the paper does not properly acknowledge the numerous instances of prior work exploring these sorts of improved loss function.

- The set of baselines is very limited.

- There does not seem to have been any proofreading of the resulting PDF, as all references are ill-formatted.

- Several of the tables are poorly formatted, with fonts squashed or stretched.

**Questions:**

- LLMs likely already possess knowledge from UMLS. Would it be possible to develop an approach that better leverages the intrinsic knowledge of LLMs instead of depending on extra training examples for each piece of knowledge in UMLS?

---

### Meta-Review · Area_Chair_cDUb · 2026-01-07

**Summary:**

In my assessment, the paper is not yet ready for publication at ICLR. All reviewers have recommended rejection, raising several significant concerns. These include unclear motivation, insufficient clarity in the presentation, limited baselines and experimental results, as well as formatting issues. Importantly, the authors did not provide a rebuttal to address these concerns, nor did they update the manuscript in response to the reviewers’ feedback. Given the lack of engagement with the review process and the unresolved issues, I do not recommend acceptance of this submission in its current form.

**Reviewer Concerns:**

None of the concerns were addressed as the authors did not provide a response.

**Reviewer Scores:**

With no response from the authors, the initial ratings are unlikely to change.

---

### Decision · Program_Chairs · 2026-01-26

Reject